# Automatic Robot-Driven 3D Reconstruction System for Chronic Wounds

**DOI:** 10.3390/s21248308

**Published:** 2021-12-12

**Authors:** Damir Filko, Domagoj Marijanović, Emmanuel Karlo Nyarko

**Affiliations:** Faculty of Electrical Engineering, Computer Science and Information Technology Osijek, Josip Juraj Strossmayer University of Osijek, HR-31000 Osijek, Croatia; domagoj.marijanovic@ferit.hr (D.M.); karlo.nyarko@ferit.hr (E.K.N.)

**Keywords:** chronic wounds, 3D reconstruction, automatic, robot, next best view

## Abstract

Chronic wounds, or wounds that are not healing properly, are a worldwide health problem that affect the global economy and population. Alongside with aging of the population, increasing obesity and diabetes patients, we can assume that costs of chronic wound healing will be even higher. Wound assessment should be fast and accurate in order to reduce the possible complications, and therefore shorten the wound healing process. Contact methods often used by medical experts have drawbacks that are easily overcome by non-contact methods like image analysis, where wound analysis is fully or partially automated. This paper describes an automatic wound recording system build upon 7 DoF robot arm with attached RGB-D camera and high precision 3D scanner. The developed system presents a novel NBV algorithm that utilizes surface-based approach based on surface point density and discontinuity detection. The system was evaluated on multiple wounds located on medical models as well as on real patents recorded in clinical medical center.

## 1. Introduction

Chronic wounds heal very slowly, and the healing process may further be prolonged if an ineffective treatment is used. Clinicians need an objective wound characterization method to decide whether the current treatment is adequate or requires modifications. Accurate wound measurement is an important task in chronic wound treatment, because changes in the physical parameters of the wound are indicators of the healing progress.

Chronic wound analysis methods can be divided into contact and non-contact methods. Contact methods such as manual planimetry using rulers, transparency tracing, color dye injection, and alginate molds, are considered traditional and have been used mostly in the past [1,2]. They are usually very painful for patients as well as unpractical for medical staff, and because of the irregularity of wound shape, they often lack accuracy and precision. The application of non-contact wound analysis has improved with the increase in computational capability of modern hardware. Moreover, advances in data analysis have led to the fast-growing use of the digital imaging approach in wound assessment. A recent overview of non-contact based chronic wound analysis can be found in [3].

Wound recording process is non-trivial, since the wound can theoretically be positioned on any part of the body and be of any size and shape. In reality, most chronic wounds that are considered here are usually located on legs such as venomous and diabetic ulcers or on a back area such as pressure ulcers. Wounds on legs usually tend to be smaller in volume while being located on a highly curved part of the body (Figure 1a), while pressure ulcers are usually located on a flatter surfaces but tend to be much larger in size (Figure 1b).

Due to the nature of chronic wounds where they can expand or reduce over the prolonged period of time, the surface geometry of those wounds can be very dynamic where some part of the wound can occlude other parts when viewed from certain directions. When reconstructing such wounds as a 3D model, the recording process can be quite complex and requiring number of stages and a multitude of recording positions. Which can be quite taxing if done manually using a hand held 3D sensor or a camera, and because of the lack of precision of human operators, the reconstructed 3D model could have a certain lack of detail at best, or abnormalities at worst [4].

Therefore, an automated system with a much higher precision than human operators and which is able to record wounds from different viewpoints would be able track the state of the recording process and enforce a specified density of surface samples on all parts of the recorded wound surface. The development of such automated system using a 7 DoF (Degrees of Freedom) robot arm and a high precision 3D scanner is the topic and the main scientific contribution of this paper. Other scientific contributions include a novel next based view algorithm that utilizes surface-based approach based on surface point density and discontinuity plane detection.

Development of a fully automatic recording system that will be able to enforce a certain surface sample density is only the first step in developing a larger system that will enable extrapolation of more concrete data such as wound’s physical parameters and percentile representation of specific tissue type. Such measurements and their progression over time will enable clinicians to track and apply appropriate therapy in a timely fashion. Furthermore, creating precise digital 3D reconstructions of patients wounds would facilitate collaboration between remote clinicians, which could result in a deeper understanding of current state of patient’s wound and better therapy proposals.

The rest of the paper is structured as follows. In Section 2 a short survey of related research is given. In Section 3, a hardware and software configuration of the developed system is described. Section 4 gives a high level overview of the developed system, while the Section 5 describes the implementation of individual components. Section 6 presents the case studies of the application of the developed system, while the paper concludes with the Section 7.

## 2. Related Research

Because of the development of increasingly sophisticated cameras and their decrease in cost, there is a lot of research and development of non-contact based wound assessment over the last decade. Only those research papers which focus on 3D reconstruction of recorded wounds will be mentioned here.

Some wound assessment systems use 3D reconstruction only to enhance image-based methods and gain more precise measurement. Therefore, using multiple view geometry algorithms with standard cameras is a typical approach. In [5], the authors use two wound images taken from different angles to generate a 3D mesh model. Because of the technology and algorithms used, the resulting 3D mesh has a low resolution.

Lasers are also commonly used for 3D reconstruction in medical research, where a laser line projection sensor calibrated with an RGB camera can produce precise and colored 3D reconstructions. One of the first studies that used such a system was Derma [6], where the authors used Minolta VI910 scanner. Other similar research also used a combination of laser and RGB cameras [7,8]. Such systems have proven to be very precise, but need to be operated manually and are therefore not easy to use and very expensive.

Filko et al. [4] considered wound detection as an important stage of wound analysis in their research. The majority of other papers assume that wound is detection is done by the user capturing the image. Therefore, apart from a Kinectfusion inspired 3D reconstruction procedure, Filko et al. [4] have a detection phase to find the center of the wound. This is implemented using color histograms and kNN algorithm. Furthermore, they also perform segmentation of the wound from the reconstructed 3D model by first performing a segmentation of the reconstructed 3D surface into surfels. These surfels are then grouped into larger smooth surfaces by a region growing process that utilizes geometry and color information for establishing relations between neighboring surfels. Finally, the wound boundary is reconstructed by spline interpolation and the wound is isolated as a separate 3D model and its circumference, area and volume is calculated.

The use of automatic robotic system for medical wound scanning is not common and application of robotics for that purpose has only started recently. 

Authors in [9] used a gantry, 2 DoF robot with a 2D camera and laser range finder combination to generate 3D point cloud for 3D reconstruction of wound edge and topology with the final purpose of developing a wound closure system. The developed system is basically indifferent in terms of what it is scanning, and would scan the entire region i.e., whole model without the automatic detection and focusing on specific area.

Authors in [10] developed a semi-autonomous robot system for trauma assessment. The developed system uses a 7 DoF robot and a RGB-D camera for creating a 3D reconstruction of a medical model. The recording process consists of semi-circular motion of the robot-mounted camera, 30 cm above the skin in order to collect point clouds and 2D images in 21 preprogramed positions. The authors also use R-CNN for classification and localization of specific regions such as umbilicus and traumas like gunshot wounds, in order to achieve better localization of FAST (Focus Assessment with Sonography for Trauma) scan positions for the use of the attached ultrasound probe.

Recent papers that use robots for scanning procedures only utilize preprogramed behaviors for recording. This could be a suboptimal use of resources and time with the patient, and does not have the ability to adapt to specific circumstances regarding the conditions on the patient’s wound surface area.

As previously mentioned, in order to create a precise 3D model of any real world surface, it is usually needed to record the surface from multiple viewpoints. There are numerous ways to generate views for the purpose of surface or object reconstruction and this general problem of generating view proposals is called next best view (NBV). 

NBV planning approaches can be categorized as either scene-model-based or scene-model-free. Model-based approaches use a-priori knowledge of the scene structure to compute a set of views from which the scene (i.e., an object or environment) is observed. These approaches work for a predefined scene but do not generalize well to all other scenes. 

Model-free approaches often use a volumetric or surface representation. Volumetric representations discretize the 3D scene into voxels and can obtain high observation coverage with a small voxel size but do not produce high resolution models of large scenes. The model resolution obtained from a volumetric representation depends on the resolution of the voxel grid and the number of potential views. Smaller voxels and more possible views allow for greater model detail but require higher computational costs to ray-trace each view. These representations are difficult to scale to large scenes without lowering the model quality or increasing the computation time. Surface-based representations estimate surface geometry from observations and can obtain high quality models of large scenes, but often require tuning of unintuitive parameters or multiple survey stages.

The system developed in this paper uses a surface based representation in order to facilitate the need for required surface point density and to reduce computational cost that would typically be required in volume representation approaches.

General purpose NBV algorithms will be discussed in the following paragraphs.

Authors in [11] use a surface-based NBV approach with a laser scanner and an industrial robot whose workspace is assigned by the coarse bounding box around the object. Firstly, the object is linearly scanned by the longest dimension of the bounding box followed by the boundary classification which separates the boundary of the object by the angle between the points edge and scan direction. For resulting boundaries, region growing is conducted starting at the vertices on the boundary which gives the estimation of the surface trend of the unknown area. Possible viewpoints for each boundary are calculated perpendicular to the estimated quadratic patch at a certain distance and amount of overlap with the previous scan. The NBV is selected using the first left boundary, and if there are no more left boundaries, it continues with right boundaries and then with top boundaries.

Research in [12] is based on the recognition of the borders of the scanned object, where a best-fit plane is calculated for those borders and the scanning device is positioned at the center of that plane along its normal vector. Models generated using this approach have a good coverage, but an increased number of measurements since the system in its final stage is filling holes in the model, resulting in a prolonged time to finish. The same authors implemented in [13], 13 state-of-the-art NBV algorithms and evaluated their performance with a robotic manipulator with 6 DoF, a rotating table, and three different 3D camera setups. The basic criteria for comparing all the algorithms were based on the number of measurements, processing time, positioning distance, and surface coverage. Tests were carried out on five different objects with different ratios of object volume and sensor working volume, (i) object smaller than working volume, (ii) same as working volume, and (iii) bigger than working volume. The overall results showed that there is no method that outperforms all other algorithms in all the working cases.

Isler et al. [14] used a mobile robot equipped with a camera to obtain a 3D model of the object. The authors generate a set of 48 candidate views around the object in a shape of a cylinder with the half-sphere on the top. The whole volume is divided into voxels where each voxel contains information gain, a metric that shows the likelihood that a voxel can be seen from a potential view. The information inside the voxel is called volumetric information and it shows how much new information can be obtained from the potential view. Information gain is calculated for each voxel using five criteria, (i) expected visible uncertainty, (ii) unobserved voxels, (iii) number of voxels on the opposite side of already observed voxels, (iv) expected visible uncertainty on the voxel’s rear side of already observed surfaces, (v) volumetric information is increased if the voxel lies close to already observed surfaces. The next best view is chosen as the view with higher volumetric information, taking into account the cost function of robot movement.

Authors in [15] used Kinect V2 depth camera together with a robot manipulator with 6 DoF for the proposed NBV algorithm called active perception system. The algorithm firstly generates starting potential views by extracting the points that lie on the border of unfinished surfaces, orthogonally to the border points. For each contour point, view direction is calculated together with four more directions in the angle range of 15°, and for each view direction, eight view poses are generated at 45° intervals at a fixed distance of 80 cm from the object, in the end resulting in 40 view poses. In the intermediate phase, the algorithm segments the point cloud into clusters, where for each segment it calculates the salient value as the segment’s roundness and the degree of isolation, with higher salient values denoting the real object of the scene. In the final phase, view poses are associated with segments and each view is evaluated by the segments salient value, i.e., the number of unknown voxels visible from that pose. The scanning procedure starts with the robot taking the first initial scan of the scene, then the algorithm computes the NBV, and if the view is reachable without collision, robot is moved to the desired position.

In the next paper from the same authors [16], a different approach is used. The system is initialized with the set of points of interest (POI) that indicates the location of the object. Around each POI, potential views are generated on a sphere of fixed radius pointing to the POI. The scene is separated with the truncated signed distance function (TSDF) into surfels that separate empty from occupied space, and frontels that separate empty from unknown space. Each potential POI view on the sphere is evaluated by a score that represents the number of unknown voxels visible from that view which is calculated from the area of visible frontels. The method takes 1.43 s to evaluate 960 potential poses with coverage of more than 90% in 8 or 9 iterations.

Newer approaches as [17] use reinforcement learning (RL) based on topological data analysis to find the NBV. RL uses topological features such as connected components, holes and shapes as information gain for its reward system and sensor poses as RL actions. For RL, a neural network is designed that takes observations as input and produces a value that corresponds to each action as output, and these values are used to decide which action to make. Three objects were used for RL training with very complex shapes consisting of holes and concave sections. Authors used Intel RealSense D415 stereo camera and improvised a 2 DOF robot manipulator in order to reduce all possible positions of the motors, and moreover, the authors divided the action space into buckets to make the RL method more efficient.

Zhang et al. [18] calculated optimal camera positions for optical coordinate measurements using a genetic algorithm with the main goal being minimization of measurement processing time while keeping the object point density optimal. The algorithm is based on the visible point analysis approach where the visibility of points is calculated with a combination of a hidden point removal algorithm and a triangulation-based intersection algorithm. Subsequently, with local optimization, a camera position is calculated that has the most visible surface points, and this position is used as starting position for global optimization of camera positions. Global optimization is performed by genetic algorithm until sufficient views are found considering the surface coverage, image overlap, and camera angles.

Authors in [19] calculated viewpoints and trajectories for aerial 3D reconstruction in outdoor environments e.g., a large building, with an RGB camera mounted on the autonomously navigated quadcopter. The algorithm is capable to calculate free and occupied space so that the aerial robot can avoid obstacles and have trajectory flight paths without collision. The user first manually defines the region of interest and several viewpoints (at safe altitude) from where the environment is coarsely scanned. From those initial views an occupancy map is computed, and from the generated map new viewpoints are calculated with an optimization technique in order to maximize total information gain while staying on the quadcopter travel budget. The 3D model is generated utilizing structure from motion and multi view stereo methods. The time required for the whole procedure is about 30 to 60 min depending on the initial views and desired quality of the scan.

Border et al. [20] developed an NBV strategy named The Surface Edge Explorer (SEE), which uses the density of measurement to separate between fully and partially observed surfaces and therefore detect surface boundaries. Measurement density is calculated with a number of neighbor points inside the specified distance, where points with sufficient neighbors are set as core, whereas others as outliners. Those points that have core and outliner points as neighbors are classified as frontier points i.e., the boundary between the surfaces. Around the frontier points, a plane is estimated and potential views are generated to observe the plane orthogonally in order to maximize information gain. NBV is selected from potential views with respect to the current sensor position and the first observation of the scene. New points from the NBV are added to the scene and reclassified as long as the algorithm performs, i.e., until there are no more frontier points left. During the process, if the frontier points are near some discontinuity or occlusion the view must be adjusted by new measurements in order to observe the total neighborhood of the point in question e.g., around the edge, or from the new view that is not occluded. In their subsequent research [21], the authors improved their algorithm with better occlusion detection, optimization of view proposals, and evaluation of point visibility. A frontier point is occluded if there are point measurements in the line of sight from the camera position to that point within a specified search distance and radius around that point. That view is marked as occluding and it is discarded. To find a non-occluding view to a target frontier point, all points inside the point’s search radius are projected onto a sphere with the target point as center. The new view proposed is the view that is farthest from any occluded sight line directions. Visibility of each frontier point from different views is presented in the visibility graph, where frontiers are connected with edges that present the point visibility, and therefore the NBV is selected as the view with the most visible new surface i.e., frontier points. Both research [20,21] are primarily implemented and tested in a virtual environment.

Mendoza et al. [22] used a 3D convolutional neural network (CNN) to directly predict the sensor position of the NBV. To train the neural network, a dataset is generated with an algorithm that reconstructs the object multiple times using different starting sensor positions and saves the calculated NBVs. The architecture of the proposed algorithm named NBV-Net consisted of three sets of convolutional and pooling layers and four fully connected layers. The output of the algorithm is one class, i.e., pose, of the possible set of poses (14 of them) that were generated on the upper half of a sphere around the object. 

NBV is also used for 3D robotic inspection systems [23] where the object view and path planning are important to reduce the inspection task. Process of Automated Inspection System (AIS) scans all areas of the object on the calculated hemispherical path and generates the color error map to rescan areas where large errors occur. Rescan strategy is performed in four steps (i) extraction phase where large error occurs, (ii) segmentation of that extracted area, (iii) view/path planning to reach that segmented area, and (iv) fusion/inspection of the segmented area, where the strategy is repeated until the whole object is inspected.

Many of the mentioned papers primarily evaluate their algorithms in virtual environment without considering real world constraints of the recording equipment that would have to achieve those recording poses. On the other hand, many real world evaluated algorithms utilize additional equipment such as spinning roundtables in order to complete their object scanning and mitigate the constraints of their used robotic platforms.

The main contributions of this paper include the development of an automatic robot driven wound recording system and the development of a real world NBV algorithm for recording process of chronic wounds from the fixed robot base position in relation to the patient. The system employs 7 DoF robot with an attached RGB-D camera and high resolution 3D scanner used in parallel, and it is based on utilizing surface point density estimation and discontinuity detection. Furthermore, the system was tested on multiple medical model wounds as well as patient wounds recorded in a clinical medical center.

## 3. Hardware and Software Configuration

The recording system hardware configuration (Figure 2) consists of:Kinova Gen3 7 DoF robot armPhotoneo PhoXi M 3D scannerLaptop computer with Windows 10 operating systemGigabit switch

The system was programed in C++ and Python. The Photoneo Phoxi 3D scanner is attached to the robot’s tool link via custom 3D printed holder. Kinova RGB camera was manually calibrated to Phoxi 3D scanner which enabled the creation of colored 3D point clouds. Phoxi 3D scanner was also manually calibrated to robot’s tool link, which made the recordings to be automatically transformed to tool reference frame and then to robot base reference frame.

## 4. Recording System Description

In this section a high level concept of the developed system will be described. The high level concept of the wound recording system can be seen on Figure 3.

Wound recording system is divided into 6 main stages:Wound detectionMoving the robot to chosen pose and recordingPoint cloud alignmentPoint cloud analysisHypothesis generation and evaluationRecording pose estimation

When the user starts the system, the user would have to manually orient the robot in the general direction of the patient and wound that is the object of recording. Following that configuration step, the purpose of the first step is to detect the wound that is located somewhere in front of the robot so that the recording process can define the surface region on which to focus the recording. For the purpose of the wound detection, the recording system captures RGB-D image pair using the integrated cameras on the Kinova Gen 3 robot arm. The RGB image is here solely used for wound detection, while the accompanied and registered depth image is used to determine the location of the wound in 3D space in front of the robot. Based on the location provided by the wound detection algorithm, the initial recording pose is determined to be at a certain distance along the surface normal from the detected wound’s central point.

The second stage controls the robot in order to achieve the desired recording pose. Furthermore, at this stage a Phoxi depth image and point cloud recordings are created, as well as new Kinova RGB image, which is then registered to Phoxi recordings. The created point cloud is then filtered so that only valid, non-zero points with color, remain. Also, if this recording is the initial recording of the reconstruction process then a follow-up wound detection is performed on the registered RGB image so that the bounding rectangle created by the wound detection algorithm is used to generate a volume-of-interest bounding box, which will be the focus of the whole recording process.

Alignment of newly acquired recordings with the ones acquired in previous recording cycles into a consistent 3D model is the role of the third stage. Depending on the current cycle, alignment can be omitted (if it is the first recording), performed by a single pairwise alignment (if it is the second recording) or can include full pose graph optimization of all recordings (third and every following recording).

Point cloud analysis is the fourth stage, and its focus is on determining recorded surface deficiencies by classifying every point in the aligned point cloud that is located within a volume-of-interest bounding box into one of four classes: core, outlier, frontier and edge. First three classes of points are used for determining surface point density, while the last one is used for detecting surface discontinuities.

The purpose of the fifth stage is to generate and evaluate a list of hypothesis that could then be used as the next best view in the recording process, resulting in the full surface reconstruction of the detected wound. A hypothesis list is filled by two sets of hypotheses related to the two types of surface deficiencies. In order to generate a first set of hypotheses, data from surface point density classes is used in a way that each frontier point generates a recording hypothesis located along its surface normal at a certain distance. These hypothesis are then clustered together using their position and orientation data into multiple distinct clusters, where centers of those clusters represent a first set of hypotheses, which are focused on deficiency formed by the sparsity of surface point samples. The second set of hypotheses are generated from surface discontinuities where edge points detected on depth image are clustered together using only their position on a 2D image. Those 2D clusters are then used to generate a 3D discontinuity planes called DPlanes using a 3D information from those clustered 2D points. All hypothesis are then evaluated by mostly considering the number of frontier points and DPlanes visible from certain hypothesis and the distance to previous recording position.

Final stage takes a sorted list of evaluated hypothesis and checks whether they are reachable by a robot arm. If the certain hypothesis is reachable, then it is chosen for recording as the next best view. If it is not reachable, then the system generates a number of adjected views to the currently considered hypothesis and checks whether they are reachable instead. The first hypothesis that is found reachable in this procedure is chosen for recording.

When there are no hypothesis for recording generated or if none of the hypothesis (or their adjacent views) are reachable, the recording process stops and the final point cloud is generated as a voxel filtered version of the point cloud comprised of all aligned recordings.

## 5. Recording System Implementation

This section discusses implementation details for each stage.

### 5.1. Wound Detection

As mentioned earlier, wound detection stage is the entry point of the recording system and it facilitates the automatic recording procedure by the initial detection of the considered wound in front of the robot so that the first recording with the high resolution (and low range) 3D scanner can be made in optimal distance from the recorded surface. Therefore, the wound detection stage uses low resolution RGB-Depth camera pair that is integrated into Kinova Gen3 robot arm itself. The robot arm provides intrinsic and extrinsic factory calibration data for the camera pair that are used for RGB to depth registration and creation of appropriate colored point cloud, which is then used for locating the wound in 3D space relative to the robot base.

The wound detection algorithm uses only RGB image and outputs the bounding rectangle for each detected wound. Wound detection was performed using a classifier based on deep convolutional neural network, MobileNetV2, with connected component labelling described in [24]. The classifier was trained using data from the Foot Ulcer Segmentation Challenge [25] as well as our own database of wound images annotated by medical experts. The output of the classifier was modified to produce a rectangular region of interest marking the wound area.

At this stage if there are multiple wounds detected, the user would need to manually select the wound that will be reconstructed and analyzed by the recording procedure. If there is only one wound detected, the system automatically chooses it and determines its 3D location and surface normal of the bounding rectangle’s image center using the recorded data. The 3D location is determined from the generated colored point cloud, while the surface normal is estimated from the point neighborhood by calculating the covariance matrix and its eigen values and vectors.

By finally choosing the focus point on wound surface and its normal in recorded 3D environment, the system determines the first recording position to be located along the surface normal at the certain distance from the chosen point. The distance from the surface point is chosen according to the optimal recording distance in the Photoneo Phoxi M specification, which is stated to be 650 mm. Figure 4a show an example of the registered RGB image with the bounding rectangle generated by the detection algorithm. The image consists of mostly black color since the image is in the resolution of depth image, which has a larger field of view than the RGB camera when recording at HD camera resolution.

Figure 4b shows a 3D visualization of current (grey color) and requested (orange color) robot positions, as well as the location and surface normal of the center point of the detected wound bounding rectangle. The current robot position in the visualization was generated from joint angles provided by the robot arm, while the requested position vas generated by the robot’s inverse kinematics API check.

After the initial wound detection stage, the Kinova power down its depth camera because it will no longer be needed since all following 3D recording will be made by Phoxi 3D scanner, but also in order for the Kinova depth camera’s IR projector to not interfere with Phoxi projector. The Kinova RGB camera is left operating since it will be used in conjunction with Phoxi for every following recording.

### 5.2. Moving Robot to Pose and Recording

Following the estimation of the initial recording pose, and every other chosen recording pose after that, the robot is instructed to move to a new pose. The robot moves in such a way that it places its tool reference frame in the chosen position and orientation. As can be seen on Figure 5, the tool reference frame is located roughly at a midpoint along the global reference frame Z axis, between Phoxi 3D scanner reference frame and Kinova’s RGB camera reference frame. Tool reference frame was chosen as the goal among other possible reference frames (Kinova RGB camera reference frame, Phoxi reference frame) in order to achieve maximal overlap between RGB and Phoxi field of views. It can also be seen on Figure 5 that the Phoxi reference frame is at an angle to the other reference frames, which is related to the work principle of the Photoneo Phoxi 3D scanners.

After Kinova robot arm acknowledges the arrival to a designated pose, the system pauses for 2 s in order for the arm to stop vibrating before creating recordings with Phoxi 3D scanner and Kinova RGB camera. RGB camera is also instructed to autofocus before making the recording. Following the grabbing of images, Kinova RGB image with resolution of 1920 × 1080 is registered to Phoxi depth image with resolution of 2064 × 1544, which leads to the creation of high resolution colored point cloud transformed to a robot’s base reference frame. That colored point cloud is then filtered so only valid, non-zero and colored points remain and is sent to alignment stage. If the current recording is the first recording of the wound then bounding box volume-of-interest also needs to be defined. For that purpose, the wound is detected for the second time on the new registered RGB/Phoxi image as can be seen on Figure 6a and the resulting bounding rectangle on the 2D image is used to create a 3D oriented bounding box that includes all 3D point cloud points that are related to their 2D image projections enveloped by 2D bounding rectangle. The resulting 3D bounding box is oriented according to the resulting eigen vectors and values from the covariance matrix calculated from included points, and its dimensions are also inflated by 10% in order to be certain the whole wound will be included in its volume. The example of the recorded color point cloud with generated bounding box in regards to the position of the robot arm can be seen on Figure 6b.

At this stage an additional data is also created in the form of two subsampled point clouds by using voxel filtering, one fine with a voxel size of 2 mm and the other coarse with 20 mm voxel size. For the coarse point cloud FPFH descriptors are also created.

### 5.3. Point Cloud Alignment

As mentioned earlier, alignment stage aims to create a coherent 3D point cloud by integrating all recordings, made up to this point in time, into one global point cloud which will be analyzed for possibility of finding next best view and in the end be used for creation of final point cloud. Alignment stage works differently based on how many recordings have been taken thus far.

Therefore, if there is only one recording, no alignment is needed. If there are two recordings, then a pairwise alignment is applied and if there are three or more recordings then a pose graph optimization is also performed.

Because of the imperfect internal measurement of the robot arm pose, point clouds recorded and transformed into robot base reference frame can be noticeably apart depending of the distance from scanner to recording surface. Therefore, a pairwise alignment between any two point clouds in this stage is performed with a coarse-to-fine approach consisting of three steps:Coarse RANSAC registration using coarse subsampled point clouds and using FPFH descriptors for matching.Coarse point-to-plane ICP registration using fine subsampled point clouds and weak constraints in the form of larger maximum corresponding distance for point pairs, which was set to 10 mm.Fine point-to-plane ICP registration using fine subsampled point clouds and tight constraints in the form of smaller maximum corresponding distance for point pairs, which was set to 5 mm.

When applying pose graph optimization, the graph consists of nodes, which are the recorded point clouds with associated global transformation matrices, and edges that represent transformation matrices calculated by pairwise alignment between pairs of nodes in the graph. The object of pose graph optimization is to find those global transformation matrices that transform recorded point clouds into the same base reference frame while minimizing the overall error of the registration. The pose graph optimization used here is implemented in Open3D library [26], which is based on research [27]. An example of aligned global point clouds made of multiple recordings can be seen in Section 6 of this paper.

### 5.4. Point Cloud Analysis

Point cloud analysis is the central stage of recording process because the data generated here will be central to determining the next best view for recording and completing the 3D wound reconstruction. The basis for determining the next best view is by detection of surface deficiencies like sparsely sampled surfaces and surface discontinuities.

The sparsity of recorded surface is determined by the counting the number of surface samples in the global point cloud over a designated area unit such as the circular area with defined radius, for example demanding a 60 surface samples in a circular area with radius of 2 mm centered at each point. The discontinuities on the other hand are detected on the depth image by measuring the distance value between neighboring pixels and requiring that it is above a certain threshold e.g., 15 mm.

In this stage, every point located in the designated bounding box volume-of-interest is labeled as one of the 4 classes: core, outlier, frontier and edge.

Edge points are the first one classified by analyzing the current depth image and only those pixels whose 3D projections are included within the defined bounding box. For each designated pixel a 8-pixel neighborhood is checked and if at least in one case the absolute distance between the central point and its neighbor is above threshold, then the central point is labeled as edge. The algorithm can be seen in the Algorithm 1 where the input variable *I_BB_* is list of indices from current recording located within the bounding box, *D* is the current depth image and *d_thr_* is the threshold value. The algorithm ends with an updated list of edge points *E*. The threshold value was set to 15 mm, which was found to be optimal in our experiments. The algorithm is also easily parallelizable, which makes it very fast. An example of detected edge points on depth image can be seen on Figure 7c, while the example of edge labeled 3D points can be seen on Figure 8a.
**Algorithm 1.** Edge Point Detection***DetectEdgePoints(****I_BB_*, *D*, *E*, *d_thr_**):****D_E_**←**Ø**N**← list of relative neighbor addresses for 8-pixel neighborhood****for****i **in** I_BB_*  *idx*
*← I_BB_[i]*  *d*
*← D[idx]*  ***for** n **in** N*    *d_n_*
*← D[idx + n]*    ***if** |d − d_n_| > d_thr_*      *D_E_*
*← add idx to list*      ***break***
***for****E ←**E + D_E_*

Core, outlier and frontier points are classified after the edge points since edge points are excluded from surface density analysis. Core, outlier and frontier points are defined as follows:Core points are all points that are not labeled edge and in their neighborhood defined by a radius contains a requested number of surface points.Outlier points are points that are not labeled edge and that do not have a requested number of points in their neighborhood.Frontier points are points that have core and outlier points in their neighborhood and represent the frontier between densely and sparsely sampled surfaces.

In concept, this algorithm is similar to the classifying algorithm in [20], but is made much simpler in order to be easily parallelizable and therefore faster for dense point clouds. The algorithm for point classification can be seen in the Algorithm 2. Where *PCD* is the aligned point cloud, *PCD_BB_* is the list of all points from the new recording located within the bounding box, *C* is the list of core points, *O* is the list of outlier points, *E* is the list of edge points, *r* is the queried neighborhood radius and *n_thr_* is the requested number of neighbor points in order for the point to be classified as core.
**Algorithm 2.** Algorithm for Classifying Points into: Core, Outlier and Frontier***ClassifyPoints(****PCD*, *PCD_BB_*, *C*, *O*, *E*, *r*, *n_thr_**):****N**← Ø**F**← Ø**PCD_BB_**←**PCD_BB_ − E****for****p **in** PCD_BB_*  *N*
*← insert neighborhood(PCD*, *p*, *r)****for****o **in** O*  *N*
*← insert neighborhood(PCD*, *o*, *r)**P**← PCD_BB_ U O****for****p **in** P*  ***if** length(N[p]) > n_thr_*    *C*
*← add p*    ***if** p **in** O*      *Remove p from O*  ***else***    ***if** p **not** in O*      *O*
*← insert p****for****p **in** O:*  ***for** p_n_ **in** N[p]:*    ***if** p_n_ **in** C **and** p_n_ **not in** F:*      *F*
*← insert p_n_*      *Remove p_n_ from C*

As it can be seen in beginning of Algorithm 2, the input points are pruned by the detected edge points since those points are already classified. Following that, point neighborhoods are calculated for the new points and for older, already classified, outlier points in order to check if they have enough new neighbors to become core points. Afterwards, the lists of new points and outlier points are grouped together for the purpose of classification into core and outlier points. Frontier point classification is done after that initial classification into core and outlier, and it is done in a way that all core points found in the neighborhood of outlier point are labeled as frontier. The algorithm ends with a list of frontier points *F* and updated lists of core points *C* and outlier points *O*, while the list of edge points *E* has been created earlier in Algorithm 1. Therefore, the lists of core, outlier and edge points are kept updated across the cycles of recording while frontier points are always newly classified for every cycle. An example of labeled core and outlier points can be seen on Figure 8b, while the addition of labeled frontier points can be seen on Figure 8c.

### 5.5. Hypothesis Generation and Evaluation

Hypothesis generation stage produces a list of possible hypotheses for the NBV, which are evaluated based on the number of surface deficiencies (sparse surfaces, discontinuities) that can be seen from those poses in order to eliminate them, as well as the distance from the current recording position in order to minimize the distance traveled by robot during the whole recording procedure.

Hypotheses are independently generated from surface density deficiencies, i.e., from frontier points and from surface discontinuities.

Hypotheses from frontier points are generated by clustering sub-hypothess that are in turn generated from individual frontier points by applying a predefined recording distance from their position along their surface normal. A full algorithm can be seen in Algorithm 3. where *PCD* is the positional data of the point cloud, *F* is the list of frontier point indices, *N* is the normal data for the point cloud, *d_rec_* is the specified recording distance, *min_c_* is the minimal number of clusters, *step_c_* is step increase in the number of clusters w.r.t number of frontier points.
**Algorithm 3.** Algorithm for Creating Recording Hypothesis from Frontier Point Data***GenerateFPointHypothesis(****PCD*, *F*, *N*, *d_rec_*, *min_c_*, *step_c_,**):****SH**← Ø**H**← Ø****for****f **in** F*  *position ← PCD[f] + d_rec_* * *N[f]*  *Z = −**N[f]*  *X* = *cross([0*, *0*, *1]*, *Z)*  *Y* = *cross(Z*, *X)*  *R* = *[X*, *Y*, *Z]*  *orientation*
*← quaterion(R)*  *SH*
*← insert (position, orientation)**no_c_**← min_c_ + length(F)/step_c_**H**← KMeans(SH*, *no_c_**)*

As can be seen from Algorithm 3, each frontier point defines a pose located along the frontier point normal at a specified recording distance and oriented toward the frontier point itself. That pose is then used for the creation of a sub-hypothesis *SH*, which is used as a clustering feature consisting of a vectors of 7 values containing pose translation and orientation as quaternion. Those features are then clustered by K-means algorithm, which has proven to be fast and accurate for our purposes with the only downside that a number of clusters need to be predefined. Therefore we employ a simple scheme for scaling the number of clusters with the number of frontier points. Following the clustering procedure, cluster centers *H* are then used as final hypothesis generated from surface density deficiencies. Figure 9 shows an example of sub-hypothesis, clustering and final hypothesis generation.

Discontinuity planes (DPlanes) are mostly generated from the following reasons:Self-occlusion by the surface geometry.Lack of depth measurement due to surface property such as reflectivity coupled with low angle view direction.

Generating hypothesis from surface discontinuities is a little more complex and the algorithm can be seen in Algorithm 4, where *H* is the hypothesis list (created in Algorithm 3), *DPlanes* is a list of discontinuity planes that is updated across the cycles, *PCD* is global aligned point cloud, *D_E_* is list of edge points detected only on the current depth image, *V* is the current recording position and *v_thr_*, *e_thr_*, *o_thr_*, *d_rec_*, *min_c_*, *step_c_*, *r*, are appropriate thresholds and parameters that will be explained in the next paragraph.
**Algorithm 4.** Algorithm for creation of DPlanes and their hypothesis***GenerateDPlaneHypothesis(****H*, *DPlanes*, *PCD*, *D_E_*, *V*, *v_thr_*, *e_thr_*, *o_thr_*, *d_rec_*, *min_c_*, *step_c_*, *r**):****edge_features**← D_E_**(x*,*y)**no_c_**← min_c_ + length(D_E_)/step_c_**C**← KMeans(edge_features*, *no_c_**)**planes ← Ø****for****c **in** C*  *pts*
*← PCD[D_E_**(c)]*  *μ*
*← CalculateMean(pts**)*  *C*
*← CalculateCovarianceMatrix(μ*, *pts)*  *e*
*← CalculateEigenValues(C)*  ***if** CheckIfDegenate(e*, *e_thr_**)*    ***continue***  *planes ← RANSAC(pts)****for****p **in** planes*  *UpdateDPlaneOverlap(p*, *PCD*, *r)*  ***if** (p.overlap/length(p.E))*
*≥ o_thr_*    ***continue***  *dot_res = dot(V*, *plane.N)*  ***if***
*|dot_res| > v_thr_*     ***if** dot_res > 0*
*p.normal = −1* * *p.N*  ***else***    *pd = duplicate(p)*    *pd.normal = −1* * *pd.N*    *DPlanes*
*← insert pd*  *DPlanes*
*← insert p****for****dp **in** DPlanes*  *position ← dp.centar + d_rec_* * *dp.N*  *Z = −**dp.N*  *X = cross([0*, *0*, *1*], *Z)*  *Y = cross(Z*, *X)*  *R = [X*, *Y*, *Z]*  *orientation*
*← quaterion(R)*  *H*
*← insert (position*, *orientation)*

As can be seen in the Algorithm 4, in the beginning, clustering features are defined as 2 element vectors containing edge point 2D coordinates from the current depth image (e.g., Figure 7c). After which, a K-means clustering is applied in order to extract 2D locally defined clusters, (e.g., Figure 7d) where the number of requested clusters is dynamically scaled with accordance to the number of edge points detected on the current depth image by the parameters *min_c_* and *step_c_*. Each cluster’s related 3D points are then separately analyzed, first, in order to analyze a dispersion of points, a mean and covariance matrix is calculated and eigen values are also calculated. If one eigen value is considerable larger (compared to a parameter *e_thr_*, e.g., more than 10 times) than the other two, than that cluster is considered to be non-planar and is discarded from further analysis. Clusters that pass that check are used for fitting a plane by using RANSAC algorithm which has proven to create a more suitable planes than by using eigen vector corresponding to smallest eigen value of the cluster’s covariance matrix. Figure 10a shows an example of generated DPanes. Each DPlane is also further analyzed by checking whether it is defined over a previously scanned surface using radius *r*, by calculating its overlap factor and comparing it to a threshold *o_thr_*. Algorithm for calculating overlap can be seen in Algorithm 5. Each DPlane normal is oriented with respect to current recording position but it is also checked whether its normal orientation is close to perpendicular to current recording orientation. If it is close to perpendicular, e.g., 80–90°, then it cannot be decided with high certainty whether to orient plane normal one way or the other. In that case the entire DPlane is duplicated so both directions are represented. Figure 10b shows duplicated and reoriented DPlanes. After the normal orientation is decided, the DPlane is fully defined and the hypothesis can be generated by placing their pose along the normal at the specified recording distance, defined by the parameter *d_rec_*, from the plane center and orienting it toward the plane. The DPlane hypothesis is added to the same list of hypothesis *H* generated by the frontier points in Algorithm 3. Figure 10c shows an example of hypothesis generated by DPlanes.

The DPlanes persist across recording cycles until the unknown surface bounded by the DPlane is scanned. Therefore the DPlanes are checked whether they are invalidated on two occasions:In the beginning of the new cycle, old DPlane’s edge points proximity are checked for the presence of the new surface samples.Newly defined DPlane’s edge point proximity are checked for the presence of the old surface samples.

The checking of old DPlanes is needed in order to find out if the unscanned parts of surfaces bounded by the discontinuity planes has been finally scanned, which makes those planes and their hypotheses unnecessary. The checking of new DPlanes is needed because by itself they are defined only on the current depth image, which could be recorded in such a way that a surface geometry self-occlude previously scanned parts of the surface that also makes those discontinuity planes unnecessary.

The purpose of those checks can best be seen on the examples in Figure 11. On Figure 11a an example of recording from a camera with a limited field of view can be seen. More closely dashed lines represent the camera view limit, while sparser dashed lines represent a ray that bounds the discontinuity on the visible surface due to self-occluding surface geometry. Thicker surface lines represent portions of the surface visible by a camera at the time of recording, while thinner lines represents unseen surfaces. On Figure 11a a DPlane designated by ① and constructed from edge points labeled in purple can also be seen. The detected DPlane also has a surface normal defined in the desired direction. An example of invalidation of old DPlane can be seen on Figure 11b where it can be seen that the camera movement to the right has resulted in a better but also incomplete view of the unseen surface beneath the ridge. That second recording has invalidated all previous edge points (here labeled blue) resulting in removal of the old DPlane ① and the creation of the new DPlane designated ② from the new detected edge points on the previously unseen parts of the surface. On the Figure 11c an example can be seen where the camera would move from the poison on Figure 11a to a position where the unseen surface bounded by the DPlane ① cannot be seen. Furthermore, all the previous edge points remains standing and the new detected edge points are instead invalidated since they are detected on a previously recorded surface. This results in invalidation of newly created DPlane ③, while the old DPlane ① remains active.

The Algorithm 5 shows how the number of overlapping edge points is updated. The algorithm works for every edge point of a certain DPlane that is not already flagged as overlapped by finding its neighborhood and checking whether the indices of points in the neighborhood belong to a certain type. If the old DPlanes are checked, then the indices are compared to the indices of the new recording, while if the new DPlanes are checked, the indices are compared to the indices of previous recordings. In the Algorithm 4, an example can be seen of how DPlane’s overlap ratio to its number of edge points is calculated and compared to a threshold (e.g., 0.95) and a decision is made whether to invalidate and delete the new DPlane. The input to the Algorithm 5 is *DPlane*, which is to be analyzed, the point cloud data *PCD* and the radius *r* for neighborhood calculation. As mentioned earlier the algorithm runs through all edge points associated to DPlane and at first they are checked whether they were already designated as overlapped some time earlier. Then the indices of the point neighborhood are fetched and their values are checked by criterion *CheckIdx*, which depends whether the considered DPlane is new or old. At the end, if the required condition is true then the edge point is flagged as overlapped and the number of overlapped points is increased.
**Algorithm 5.** An algorithm for updating DPlane’s overlap***UpdateDPlaneOverlap(****DPlane*, *PCD*, *r**):******for****e **in** DPlane.E*  ***if***
*e.overlap_flag*    ***continue***  *indices = GetPointNeighborhood(e*, *PCD*, *r)*  ***for***
*idx **in** indices*    ***if***
*CheckIdx(idx)*      *DPlane.overlap*
*← DPlane.overlap + 1*      *e.overlap_flag*
*← True*      ***break***

In the end, the list containing hypotheses generated from both frontier points and DPlanes are evaluated and their score is calculated. Each hypothesis is evaluated by the following expression:(1)$$h_{score}=\alpha \times N_{v}+\beta \times e^{-2*d}+\gamma \times N_{h},$$
where *α*, *β* and *γ* are weights in order to scale contributions from the number of visible points, distance to previous recording position and hypothesis size. During testing, weights *α*, *β* and *γ* had values of 0.8, 100 and 0.2 respectively. *N_v_* is the number of points visible from the hypothesis pose, visible points include the number of frontier points and DPlane’s edge points visible from the hypothesis pose. The check whether the frontier point or DPlane is visible, is performed by finding out if it is within a view frustum of the simulated camera positioned at the hypothesis pose. Also by further calculating the dot product of the frontier point/DPlane normal and the orientation vector, which is the z-axis of the hypothesis pose rotation matrix, and checking whether the resulting *cosϕ* is below a certain threshold. Each frontier point found visible adds 1 to the *N_v_*, while each DPlane adds the value corresponding to the number of edge points it consists of. Distance to the previous recording position is *d*, while *N_h_* is the hypothesis size, which is equal to the number of frontier points in a cluster or the number of edge points in the DPlane that produced the hypothesis. Following the evaluation, the hypothesis list is sorted based on the achieved score in a descending fashion and is further evaluated in the next stage where the check is performed to see whether the hypotheses are actually reachable by the robot arm.

### 5.6. Recording Position Estimation

Once the hypotheses are crated and evaluated based on score, further checks need to be performed before ordering the robot to move to the new pose. While usually there are several hypotheses or even dozens of them, which are evaluated with very high scores, there is no guarantee that they are actually reachable by the robot arm from its current fixed position relative to the patient. Therefore, in this stage the hypothesis from the sorted list are analyzed in order to check whether they are reachable, and if they are not, then to see if the new hypothesis can be generated in its vicinity that is reachable.

In order to determine appropriate next best view that will be used for the recording, this stage has several substages that can be seen in Figure 12.

The input hypothesis from the sorted list are analyzed from the highest rated to the lowest, and the first one that is reachable (or produces a nearby pose that is reachable) is used as the next recording position. 

First, a proximity check is performed in order to avoid a new recording position close to any previous recording positions. This is to avoid a recording system getting caught in a loop where it tries repeatedly to scan a certain wound surface but because of the surface geometry and the position of cameras on the robot, it cannot record the required surface from the requested position.

The robot pose reachability is checked by using Bullet library [28] for simulating the robot arm forward kinematics in the physics based 3D space and checking whether the pose reached by the tool reference frame of the 3D model is close to the requested pose by the hypothesis. The inverse kinematics is calculated by the Robotics Toolbox for Python library [29], which was found to be more precise than inverse kinematics implemented in the Bullet library. The 3D model of the robot arm used for physics simulation was put together from official 3D models provided by the manufacturer of the robot arm and 3D scanner, as well as our own 3D model for holder that connects 3D scanner to the tool link of the robot arm. The reachable state of the robot arm is found true if the difference between the requested pose and achieved position is within the tolerance and if the angle difference of the orientation is also within the tolerance. The position difference is easily calculated by the Euclidian distance, while the angle difference was determined by finding the norm of the rotation vector, which was calculated from the result of matrix multiplication of the two rotation matrices where the hypothesis rotation matrix was transposed.

As can be seen in the Figure 12, if the physics-based checker finds the pose reachable, the hypothesis is accepted and the robot is instructed to move to the requested pose. If the pose is not reachable, the algorithm tries to find a nearby pose that is reachable by generating a series of poses arranged in a sphere slice around the requested hypothesis and oriented toward the point the original hypothesis was oriented to as well. An example of that sphere generation of nearby hypothesis can be seen in Figure 13a.

After the generation of sphere nearby poses, they are also evaluated in a similar way as the normal hypothesis. They are firstly evaluated by score according to expression (1), except where the part related to the hypothesis size in the expression is 0 since these hypotheses weren’t generated by clustering, so there is no size assigned to them. After they have been evaluated by score, they are sorted and checked by a physics-based reachability checker. Since there could possibly be several hundred of the sphere poses to check, in order to speed up the checking, the checker is run in a number of separate process in parallel. The reachability check of the sphere poses are run until the first reachable pose is found, and since the poses are sorted by score, the found reachable pose is also the best one available. Figure 13b shows reachable sphere poses if left to check every pose instead of finding only the first reachable one, while Figure 13c shows the new robot arm configuration orange colored in the highest rated reachable sphere pose.

In the end if the reachable pose is found among the near poses generated in a sphere pattern, the system continues with the recording procedure, and if it doesn’t find anything, then it continues with the next hypothesis in the sorted list until it finds a reachable pose. When next recording position is not found due to the current state of the reconstructed surface being in satisfactory condition where no hypothesis is generated or that no reachable pose is found, the system finishes the recording process. The system then outputs the final 3D model generated by subsampling using voxel filtering with voxel size of 1mm. The example of such 3D models can be found in the next section.

### 5.7. Reconstructed Point Cloud Precision

In order to determine the precision of the 3D reconstructions made by the developed system, a distance metric can be computed between the reconstructed point cloud of the wound and the ground truth 3D mesh model. The ground truth for assessing the precision of the proposed system was created by digitalization of the whole Saymour II wound care model using an industrial 3D scanning equipment based on GOM ATOS Triple Scan, which has a precision in 10 μm range. Glossy surfaces were treated by spraying dulling powder prior to scanning in order to remove reflectivity, an action that drastically increases the precision of the scanning process, but also an action that cannot be applied to real world patient’s wounds. Reconstructed point cloud to ground truth mesh distance was calculated using CloudCompare software [30] where prior to distance calculation point cloud was automatically aligned using the CloudCompare internal ICP alignment procedure. The point cloud used for comparison was generated from 4 views as described in Section 6.2. Aligned wound point cloud to ground truth mesh can be seen in Figure 14a. The result of point cloud to mesh distance calculation can be seen in Table 1 and Figure 14b. As can be seen, the point cloud to mesh distance calculation has yielded statistical data with a mean error of 0.14 mm and standard deviation of 0.12 mm, resulting that approximately 64% of points had an error of less ten 0.15 mm and almost 99% of points had an error less than 0.5 mm. Consequently, assessment of reconstructed point cloud precision has shown more than adequate results needed for further wound analysis.

## 6. Case Studies

In this section the recording process of two artificial wound models and two genuine wounds recorded in hospital environment is presented.

### 6.1. Venous Ulcer Medical Wound Model

Venous ulcer recorded here is located on the Vinnie venous insufficiency leg model made by VATA Inc. (Canby, OR, USA) The wound itself is relatively small and shallow, which facilitates 3D reconstruction by recording from single view. In the Figure 15a a recording process can be seen at the stage of wound detection, after the robot has moved to the first recording position; wound detection at this stage is for the purpose of defining volume of interest via bounding box. The wound detection at that stage was successful although the rectangle defined by the detection system doesn’t encompass the whole wound, the 3D bounding box is automatically inflated by a fixed percentile, as explained in the Section 5.2, in order to be certain that the recorded wounds are completely enveloped by the bounding box. On the Figure 15b,c a recording position with and without the robot model can be seen. The model of the robot is generated on the scene based on the achieved joint angle measurements provided by the robot arm at the recording position. Since the recorded wound is shallow only one recording was necessary for full reconstruction since no frontier points or DPlanes were detected. Fully reconstructed and voxel filtered model can be seen on Figure 15d, while the rendering with normal map is shown on Figure 15e in order to better distinguish the surface geometry.

### 6.2. Stage IV Pressure Ulcer Medical Wound Model

Stage IV pressure ulcer recorded here is located on the Seymour II wound care model made by VATA Inc. The pressure ulcer considered here is very large in surface area and volume, and it is also very deep with two enclosed regions made by undermining and tunneling. Because of all that, this wound is very demanding on the recording and reconstruction process, which requires multiple recording positions, and even then the tunneling parts of the wound cannot be recorded with available hardware. Result of the detection can be seen on the Figure 16a, and it can be seen that it encompasses almost the entire wound with only the small part on the bottom of the wound that is out of the resulting rectangle. Figure 16b shows all 4 automatically determined recording positions in relation to the final wound model, while Figure 16c–f shows the robot joint positions achieved during those four recordings. The final filtered wound model and rendering with normal map can be seen on Figure 16g,h, respectively. As it can be seen the wound is successfully reconstructed except for the two tunneling parts due to self-occlusion of other parts of the wound model. Since the robot couldn’t achieve the poses related to hypothesis generated by the DPlanes constructed over those orifices, the recording process finished with only 4 achievable recording poses.

### 6.3. Venous Ulcer 1 Hospital Recording

The first venous ulcer considered here that was recorded at the clinical medical center is rather small, but due to its depth, required two recording positions to fully reconstruct. Also, since the wound surface predominantly contained necrotic tissue, the wound was relatively dark and lacked visual features. As can be seen on Figure 17a, the detection system has successfully generated a bounding rectangle that encompasses the whole wound. Figure 17b shows the recording positions, while Figure 17c,d shows the robot joint positions achieved while making the recordings. As can be seen, the distance between the robot base and the wound was larger because of the safety reasons when recording the patients, as opposed to recording the artificial wound models described in previous subsections. Final wound model reconstruction can be seen on Figure 17e, and rendering with normal mapping can be seen on Figure 17f. As previously mentioned, the recording process required two recording positions since the wound was rather deep as can be seen on the Figure 17f, and the top edge of the wound wasn’t properly sampled from the first recording, which the system detected and produced a reachable recording pose. The recording process stopped when no further frontier points or DPlanes were generated.

### 6.4. Venous Ulcer 2 Hospital Recording

The second venous ulcer considered here that was recorded at the clinical medical center is rather large as it encompasses a large section of the patient’s lower leg. The wound is relatively shallow, but since it encompasses a large area, two recording positions were required in order to have it fully scanned. As it can be seen on Figure 18a, the detection system has been successful in encompassing the whole wound as it was seen from the first recording position. Figure 18b shows chosen recording positions, while Figure 18c,d shows robot joint positions while making the recordings. Final point cloud of the 3D reconstruction can be seen on Figure 18e, while Figure 18f shows rendering with the normal map.

## 7. Conclusions

Intelligent robot systems that require little or no outside input in order to do their intelligent or menial work is the near future of many fields of our society, including medicine. The lack of qualified medical staff and the increase of chronic wounds facilitated by the modern way of life will increase the adoption of robotics in the medical field. Development of fully automated medical stations for chronic wound analysis will enable gathering of objective measurements about the status of the chronic wounds such as circumference, area, volume, tissue representation, etc. The first step in obtaining those measurements in an automated fashion is the development of a robust robot system containing necessary hardware and software components needed for the precise 3D reconstruction of wound surface area. 

The system developed here enables intelligent recording pose selection based on the detection of the certain deficient surface geometry features on the scanned wound surface. Those deficient surface geometric features are based on surface samples’ density and discontinuity detection. The demand for a certain surface point density enables the detection of surface areas that are insufficiently sampled and can possibly lack details required for in-depth wound analysis. The surface discontinuity detection achieves in recognizing surface areas containing self-occlusion due to more varied surface geometry such as edges or tunnels, or simply the areas that the scanner couldn’t register due to improper surface to camera view angles. After surface geometry features are analyzed, new hypotheses for recording poses are generated with the main goal of eliminating those surface deficiencies. Hypotheses are evaluated by score, which is primarily influenced by the number of those geometric features visible from the pose that the hypothesis proposes and by the distance from the previous recording position. Furthermore the hypotheses are also checked on whether they are actually reachable by a robot arm by employing a physics-based 3D simulator. The risks of using the developed system inside hospital environment is minimal since the employed robot manipulator have torque sensors in every joint. The robot can therefore be programmed to stop at any contact with its surroundings; also there is an emergency button available. The used 3D scanner should also present minimal risk to the patient since it is using a very low power laser for its projector.

The robot reachability is the major hurdle of the developed system since the robot has a limited amount of recording poses it can achieve from the position stationary to the recording wound. This is especially troublesome if the wound has specific geometric features like tunneling, which prevents the reconstruction of the whole wound. Therefore, in the future the system would have to be expanded in order to support multiple recording sessions in order to stich models generated more multiple robot base positions. Furthermore, because of the imperfect registration between Phoxi 3D scanner and RGB camera, a color map optimization algorithm would have to be implemented in order to increase color sharpness of the reconstructed point cloud.

## Figures and Tables

**Figure 1 sensors-21-08308-f001:**
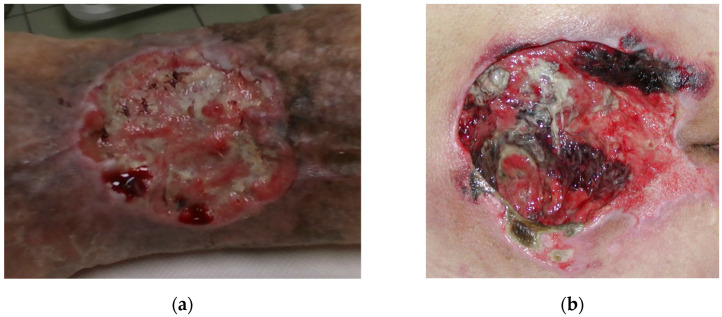
Wound located on: (**a**) leg region; (**b**) lower back region.

**Figure 2 sensors-21-08308-f002:**
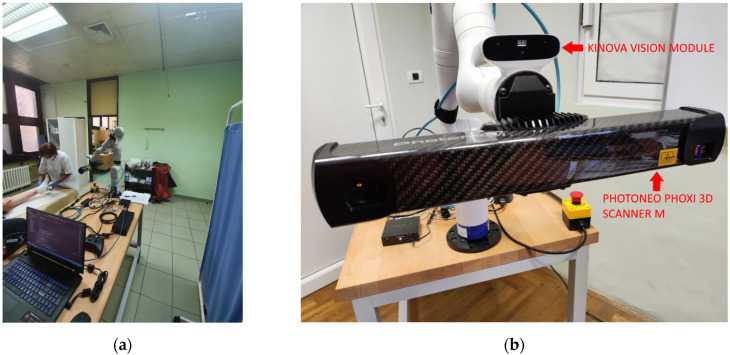
Recording system configuration: (**a**) Robot recording system in hospital environment; (**b**) cameras used in the recording system.

**Figure 3 sensors-21-08308-f003:**
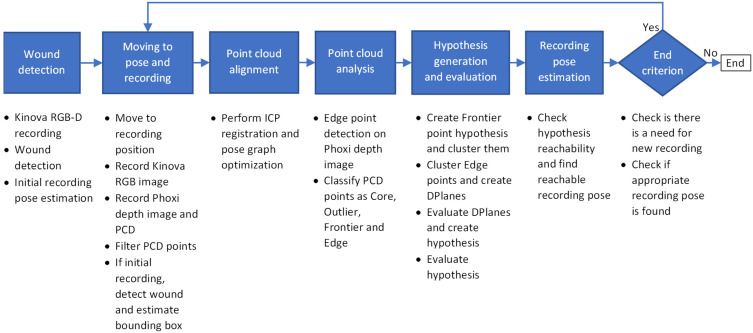
High level concept of the wound recording system.

**Figure 4 sensors-21-08308-f004:**
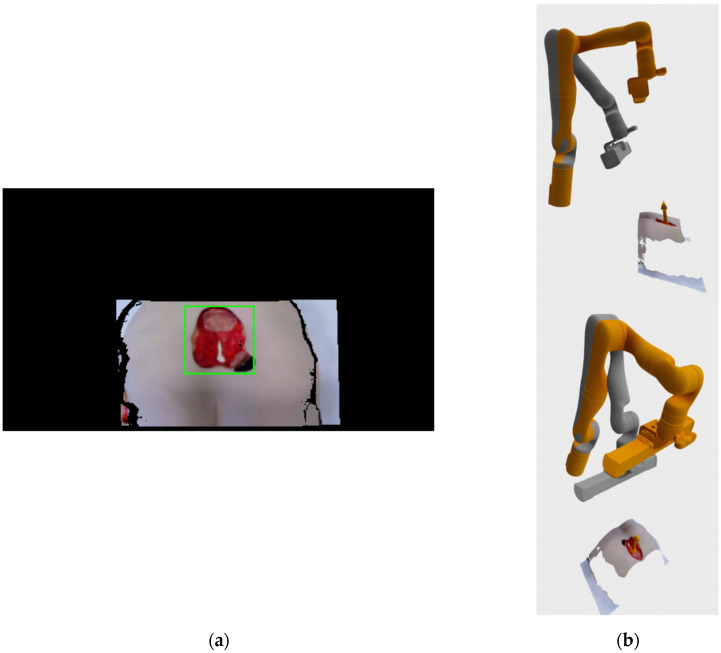
(**a**) Registered RGB-D image with bounding rectangle generated by the detection system; (**b**) 3D visualization of the current (grey) and requested (orange) robot position.

**Figure 5 sensors-21-08308-f005:**
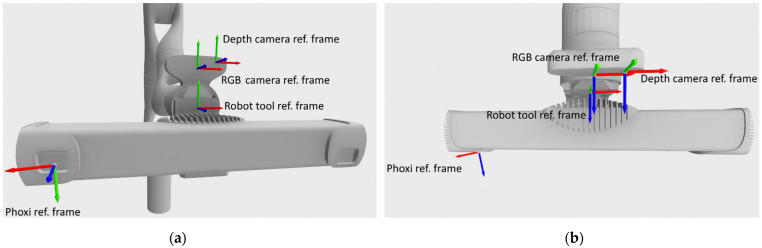
Available reference frames (X, Y, Z—red, green, blue) which could be used for robot goal movement, when viewed from: (**a**) an angle, (**b**) topside.

**Figure 6 sensors-21-08308-f006:**
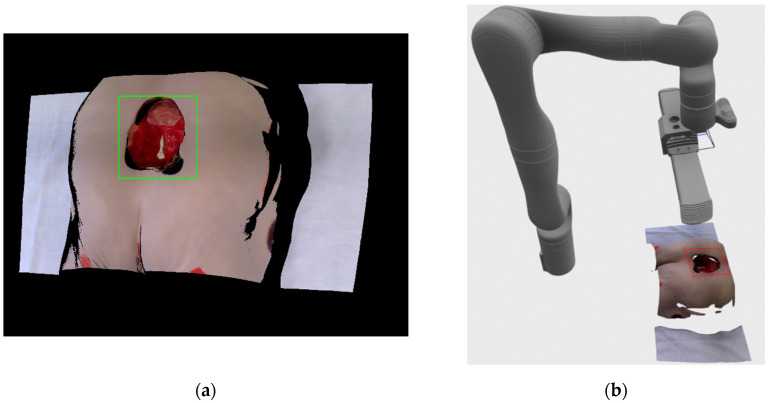
(**a**) Result of the wound detection on the registered RGB/Phoxi image; (**b**) recorded color point cloud with generated bounding box in regard to the position of the robot arm.

**Figure 7 sensors-21-08308-f007:**
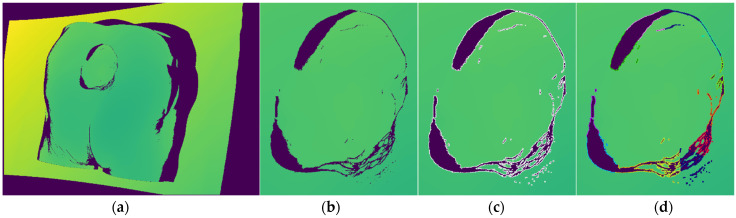
(**a**) Depth image of the medical model recorded by Photoneo Phoxi M; (**b**) close-up view of the wound area as enveloped by the detection algorithm bounding rectangle; (**c**) close-up view with labeled edge points; (**d**) clustered edge points.

**Figure 8 sensors-21-08308-f008:**
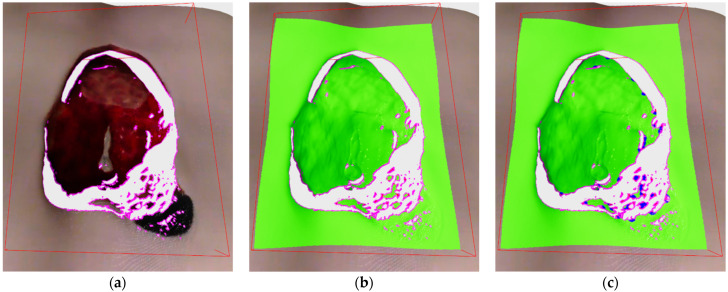
(**a**) Classified edge points in magenta color; (**b**) classified core and outlier points in green and red color; (**c**) classified frontier points in blue color.

**Figure 9 sensors-21-08308-f009:**
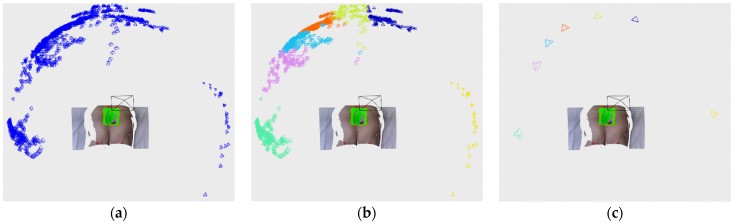
Black cone always represents the current recording position, while cones of other color represent hypothesis belonging to individual clusters: (**a**) Frontier point sub-hypothesis; (**b**) labeled frontier point sub-hypothesis according to cluster membership; (**c**) final frontier point hypothesis corresponding to cluster centers.

**Figure 10 sensors-21-08308-f010:**
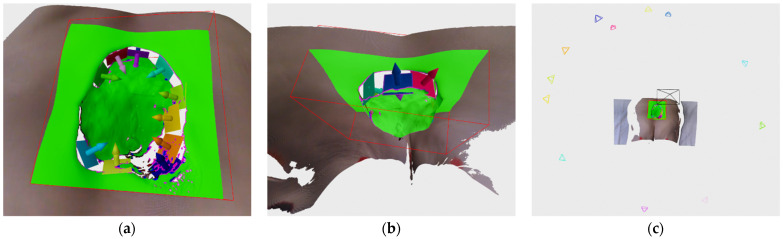
(**a**) DPlanes generated based on edge points represented by differently colored planes and normals; (**b**) duplicate DPlanes because of the plane orientation w.r.t. recording orientation; (**c**) DPlane hypothesis represented by different color cones, while black cone is the current recording pose.

**Figure 11 sensors-21-08308-f011:**
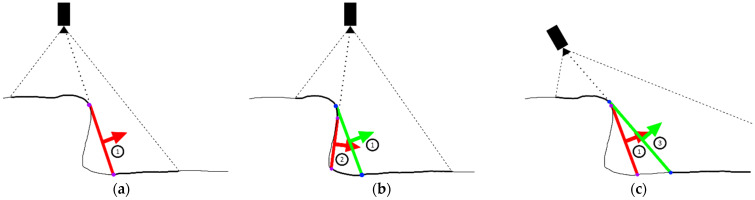
An examples of creation and deletion of DPlanes due to overlap, red colored DPlanes remain active while green DPlanes are invalidated: (**a**) Example 1; (**b**) Example 2; (**c**) Example 3.

**Figure 12 sensors-21-08308-f012:**
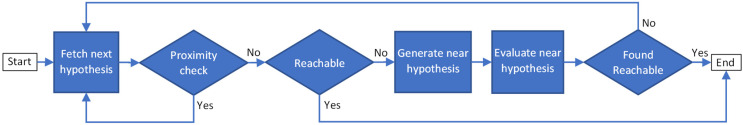
A flow chart of recording position estimation stage.

**Figure 13 sensors-21-08308-f013:**
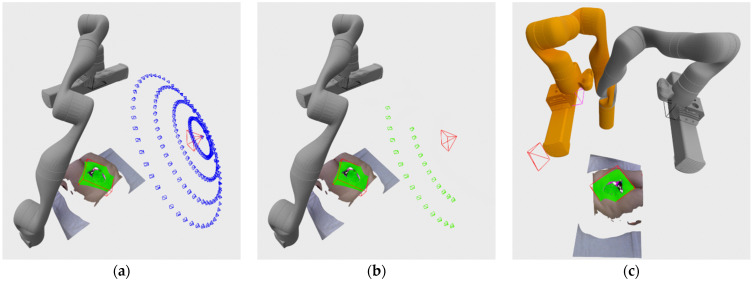
(**a**) Nearby hypothesis generated in a sphere pattern (blue) around the requested pose (red); (**b**) reachable nearby poses represented as green cones; (**c**) robot arm configuration in the nearby pose chosen as the NBV (orange).

**Figure 14 sensors-21-08308-f014:**
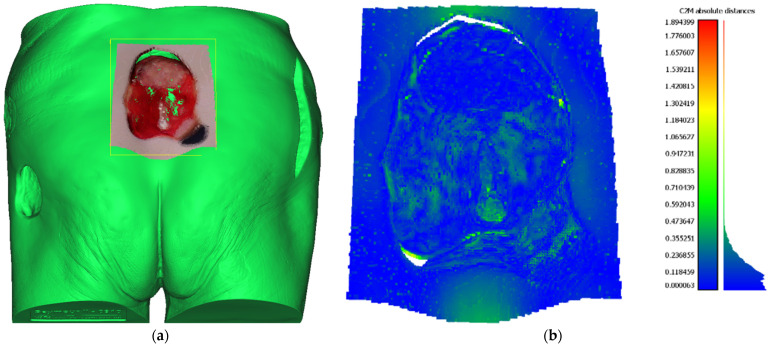
(**a**) Reconstructed wound point cloud aligned to high precision ground truth mesh (green); (**b**) Result of the point cloud to mesh distance measurement, where each point is colored according to calculated error value and in accordance with the color bar on the right side of the figure.

**Figure 15 sensors-21-08308-f015:**
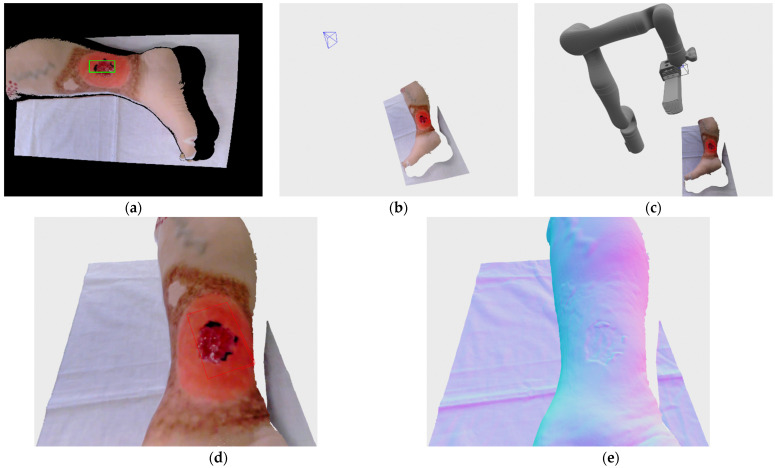
Recording of venous ulcer medical wound model: (**a**) wound detection at initial recording position; (**b**) recording pose relative to the recorded wound; (**c**) robot position in relation to the recorded wound in first and only recording; (**d**) final reconstructed 3D model; (**e**) final reconstructed 3D model with normal map rendering.

**Figure 16 sensors-21-08308-f016:**
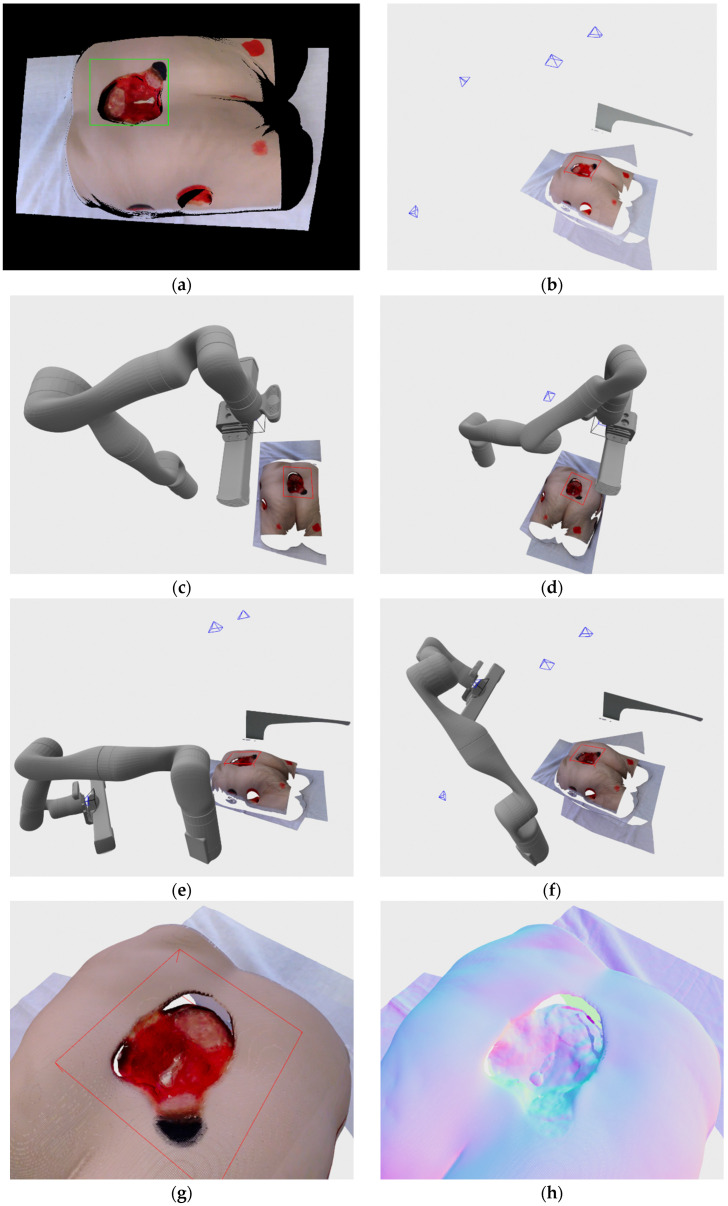
Recording of stage IV pressure ulcer medical wound model: (**a**) wound detection at initial recording position; (**b**) recording poses relative to the recorded wound; (**c**) robot position in relation to the recorded wound in first recording; (**d**) robot position in relation to the recorded wound in second recording; (**e**) robot position in relation to the recorded wound in third recording; (**f**) robot position in relation to the recorded wound in fourth recording; (**g**) final reconstructed 3D model; (**h**) final reconstructed 3D model with normal map rendering.

**Figure 17 sensors-21-08308-f017:**
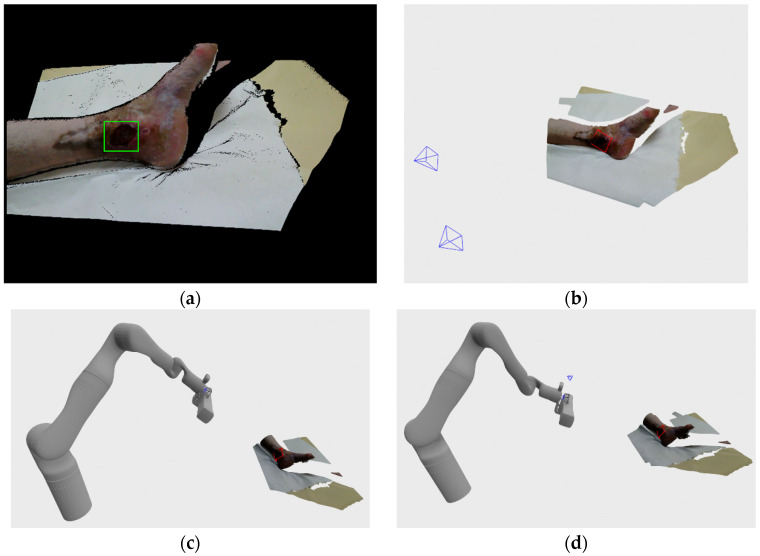
Hospital Recording of venous ulcer 1: (**a**) wound detection at initial recording position; (**b**) recording poses relative to the recorded wound; (**c**) robot position in relation to the recorded wound in first recording; (**d**) robot position in relation to the recorded wound in second recording; (**e**) final 3D model; (**f**) final 3D model with normal map rendering.

**Figure 18 sensors-21-08308-f018:**
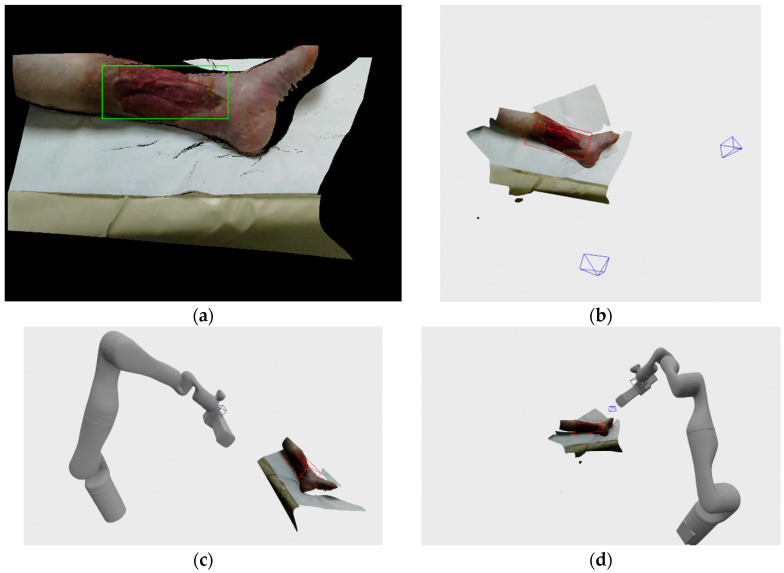
Hospital Recording of venous ulcer 2; (**a**) wound detection at initial recording position, (**b**) recording poses relative to the recorded wound, (**c**) robot position in relation to the recorded wound in first recording, (**d**) robot position in relation to the recorded wound in second recording, (**e**) final reconstructed 3D model, (**f**) final reconstructed 3D model with normal map rendering.

**Table 1 sensors-21-08308-t001:** Result of point cloud to mesh absolute distance calculation.

No. Points	Mean Error	Std. dev.	% <0.15 mm	% <0.25 mm	% <0.5 mm
21,718	0.1439 mm	0.12 mm	63.98%	86.92%	98.75%

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
