# Peer review of "Automatic Robot-Driven 3D Reconstruction System for Chronic Wounds"

_sensors, 2021, doi:10.3390/s21248308_

Round 1

Reviewer 1 Report

Filko et al report on a robotic 3D scan approach for recording of wounds. It is hard to evaluate any capabilities of the current setup as the authors provide only 4 scans of different wounds as a "result" without any scientific data, data analysis and numerical description of the systems precision based on operational parameters. How does limb positioning affect the read-out? Whats the precision of the system? whats the actual failure rate and what wounds based on what parameters can not be scanned? Does a single wound scanned 10 times yield 100% data accuracy? When stating that operator precision is a problem would be nice to compare operator reconstructions with the robot. Would be also nice to show that the system can analyze wound progression - this was the whole rationale of developing the system.

Related research section needs language revision. 'authors in xy' or 'Research in xy'? 'et al.' should be italic

Author Response

Filko et al report on a robotic 3D scan approach for recording of wounds.

It is hard to evaluate any capabilities of the current setup as the authors provide only 4 scans of different wounds as a "result" without any scientific data, data analysis and numerical description of the systems precision based on operational parameters.

We agree that there is a lack of numerical confirmation about the precision of the reconstructed wounds. It was unintentional oversight on our part as we count on the precision and reliability of the 3D scanner used in our robot setup. Therefore we included a precision evaluation in the form of the reconstructed point cloud to ground truth mesh distance metric in a new subsection 5.7.. Further discussion regarding precision is contained in following comments.

Furthermore, regarding the “only 4 scans of different wounds”, we have dozen more wounds that we scanned automatically with our developed system but showing them all wouldn’t contribute any further knowledge except showing that the system works. Therefore we chose the option of showing only typical representatives of the wounds we reconstructed. Additionally, because of the COVID-19 pandemic the majority of the clinical medical center area where we tested our developed system, where the “laying” patients are, is still inaccessible. Therefore we only had access to the patients coming to the hospital for check-up and wound dressing which had primarily venous ulcer wounds. Such wounds have rather simple surface geometry and are rather small in volume (compared to pressure ulcers for example) which results in small number of recording positions required for full reconstruction as can be seen in 3rd and 4th case study example. Therefore, the ability to reconstruct complex wounds was only tested on the synthetic wound models as stated in the manuscript, and on the most complex wound available we have provided new precision data contained in the new subsection 5.7.

How does limb positioning affect the read-out?

The limb position only affect the recording process in a way that it should enable visibility of the recording wound. Therefore the patient must be positioned in such a way that the wound is visible from as much viewing positions as possible while keeping the patient in as most comfortable position possible. Since the robot is also on a movable platform, it permits finding as optimal position as possible w.r.t. robot reachability and patient comfort.

Whats the precision of the system?

As stated in a first comment, in this revised version of the manuscript we included a precision evaluation in the form of the measuring reconstructed point cloud to ground truth mesh distance. In our prior research (ref. [4] in the manuscript) we used an outside contractor to digitize the Saymour II, wound care model using an industrial 3D scanning equipment based on GOM ATOS Triple Scan which has a precision in 10 μm range. For that ground truth scanning, glossy surfaces (entire wound surface area) were treated by spraying dulling powder prior to scanning in order to remove reflectivity, which is the process that enables a much higher precision of scanning but it is also something we cannot apply to patients. The point cloud to mesh distance calculation has yielded statistical data with a mean point cloud error of 0.14 mm and standard deviation of 0.12 mm, resulting that approximately 64% of points had an error of less ten 0.15 mm and almost 99% of points had an error less than 0.5 mm. Further results as well as visual representation of calculated distance can be seen in new subsection 5.7.

Whats the actual failure rate and what wounds based on what parameters can not be scanned?

The system is primarily designed with chronic wounds in mind, therefore wounds like pressure or venous ulcer was majority of wounds the system was tested on. The failure state in context of the developed system can be considered as a state where the wound was not fully reconstructed. As already described in the manuscript, that only happens if there are tunnelling/undermining of which we only have 1 (synthetic) example. All other considered wounds, synthetic and those recorded from patients were fully reconstructed, therefore we did not provide a full failure rate analysis and only described hardware limitations of the developed system which prevents recording of such geometric aberrations as undermining/tunneling.

Regarding parameters, in the manuscript we have stated parameter values for individual algorithms in the subsections where they are described.

Does a single wound scanned 10 times yield 100% data accuracy?

At the present time we have not detected any discernable deviations from scanning results of individual wounds that we have scanned many times (synthetic wounds). Due to COVID-19 pandemic, all wounds we have recorded at the clinical medical center we were able to only scan once and all of them were successful in terms that all wound parts were scanned and successfully integrated in a unified 3D model.

When stating that operator precision is a problem would be nice to compare operator reconstructions with the robot.

In our previous research:

Filko, D.; Cupec, R.; Nyarko, E.K. Wound measurement by RGB-D camera. Mach. Vis. Appl. 2018, 29, 633–654.

We have successfully implemented an operator driven reconstruction system that as a result, beside 3D model, outputs wound physical parameters such as circumference, area and volume.

The developed system presented in the current manuscript is the result of the first 1-2 years of development out of 5 years that the project lasts. The measuring of physical parameters of wounds will come in next years of the project and we will compare the results to an operator driven system.

Would be also nice to show that the system can analyze wound progression - this was the whole rationale of developing the system.

See previous comment.

The system is designed with the ability to analyze wound progression in mind by comparing measurements such as wound circumference, area, volume and tissue representations (granulation, necrosis, fibrin). The development of subsystems that enable those measurements will come in the later years of the project

Related research section needs language revision. 'authors in xy' or 'Research in xy'? 'et al.' should be italic

Related research section was corrected by a native English speaker, furthermore 'et al.' has been changed to italic in the whole manuscript. The 'authors in xy' or 'research in xy' was added in that section on order to prevent repetition of single statement.

Reviewer 2 Report

General comments

I would like to thank the authors for this interesting manuscript titled “Automatic robot driven 3D reconstruction system for chronic 2 wounds”.

In general, it is an acceptable article, but it needs minor changes for its publication.

1.Introduction

Line 45: I recommend citing the figure in parentheses in the text.

Line 55: place point behind reference 4.

2.Related research

Line 75: I recommend citing the authors and putting the reference at the end of the paragraph.

Lines 84-87: I recommend removing the references and placing it at the end of the paragraph.

 Line 97, 102, 137, 148, 161, 173…206…: change references and placing it at the end of the paragraph.

  1. Hardware and software configuration

Line 289-290: I recommend only citing the figure in parentheses. The explanation of the figures is already below the figure.

  1. Recording system description

Nothing to add

  1. Recording system implementation

I recommend changing figure 4a to a sharper one.

  1. Case studies

Nothing to add.

  1. Conclusions

Please add if there are risks of this method.

I think that the manuscript is complete and very well written. Good luck in the publishing process.

Author Response

General comments

I would like to thank the authors for this interesting manuscript titled “Automatic robot driven 3D reconstruction system for chronic wounds”.

In general, it is an acceptable article, but it needs minor changes for its publication.

Thank you very much for your positive opinion.

1.Introduction

Line 45: I recommend citing the figure in parentheses in the text.

Line 55: place point behind reference 4.

Thank you for the suggestions, we have applied both changes.

2.Related research

Line 75: I recommend citing the authors and putting the reference at the end of the paragraph.

Lines 84-87: I recommend removing the references and placing it at the end of the paragraph.

 Line 97, 102, 137, 148, 161, 173…206…: change references and placing it at the end of the paragraph.

Changing the position of the reference would require a rewrite of a lot of sentences since they are written in a way that they reference something that was mentioned prior to that point e.g.:

“In [5], the authors use two wound images taken from different angles to generate a 3D mesh model. Because of the technology and algorithms used, the resulting 3D mesh has a low resolution.”

The last sentence in that paragraph only makes sense if the article was mentioned and referenced in the first sentence. Therefore it wouldn’t be thoughtful if we put reference only at the end of paragraphs.

Consequently we will omit these changes for now, but if You really think we should make those changes, please mention them in the follow up review and we will make the requested changes in the final version of the manuscript.

  1. Hardware and software configuration

Line 289-290: I recommend only citing the figure in parentheses. The explanation of the figures is already below the figure.

Thank you for the suggestion, we have removed both sentences that referenced the figures and inserted the figure reference inside parentheses in the first sentence of this section.

  1. Recording system description

Nothing to add

Thank you for the comment.

  1. Recording system implementation

I recommend changing figure 4a to a sharper one.

When viewing that image in the original Word document it looks fine, but when viewing in generated PDF document it looks a little blurry. Hopefully the production staff at MDPI will be able to create a production version of the PDF based on source file with more details.

  1. Case studies

Nothing to add.

Thank you for the comment.

  1. Conclusions

Please add if there are risks of this method.

The risks of using the developed system in hospital environment is minimal and as per suggestion we have added following sentences to the conclusion:

“The risks of using the developed system inside hospital environment is minimal since the employed robot manipulator have torque sensors in every joint. Robot can therefore be programmed to stop at any contact with its surroundings, also there is an emergency button available. The used 3D scanner should also present minimal risk to the patient since it is using a very low power laser for its projector.”

I think that the manuscript is complete and very well written. Good luck in the publishing process.

Thank you very much for your positive opinion.

Reviewer 3 Report

The manuscript describes the proof of concept and experimental application of an automatic wound recording system built upon 7 DoF robot arm with attached RGB-D camera and high precision 3D scanner. The idea is interesting, and overall, the manuscript and experiments were properly conducted.

The major minus of this manuscript is related to the fact that the novelty elements are elusive. A better presentation of the new elements brought by this approach is welcomed.

Also, many similar teams are working for the automatic robotic treatment of such tissue defects, not only in examining/reconstructing them. Would you please offer a better idea in the introduction why the scanning and reconstruction of wound would present a significant improvement in the therapy of such chronic diseases?

Author Response

The manuscript describes the proof of concept and experimental application of an automatic wound recording system built upon 7 DoF robot arm with attached RGB-D camera and high precision 3D scanner. The idea is interesting, and overall, the manuscript and experiments were properly conducted.

Thank you very much for your positive opinion.

The major minus of this manuscript is related to the fact that the novelty elements are elusive. A better presentation of the new elements brought by this approach is welcomed.

Well the novelty of this research is in developing a full automated robot driven wound analysis system that would enable measurement of wound parameters such as circumference, area, volume and tissue representation. The first major part of that research is in developing fully autonomous recording platform as stated in conclusion section of the manuscript. The developed system presented in current manuscript is the first major milestone and the result of the first 1-2 years of development out of 5 years that the project lasts. The measuring of physical parameters of wounds and tissue representation by classification will come in next years of the project and more comparisons to other research results and measurements will be possible.

Even though scientific contributions of this manuscript is already described at the end of “Related research” section, in order to more pronounce scientific contributions of the presented research, we have changed and added additional information in the introduction section:

“The development of such automated system using a 7DoF (Degrees of Freedom) robot arm and a high precision 3D scanner is the topic and the main scientific contribution of this paper. Other scientific contribution include a novel next based view algorithm that utilizes surface-based approach based on surface point density and discontinuity plane detection.”

Furthermore we have also added a new subsection 5.7. where we discuss the precision of the system.

Also, many similar teams are working for the automatic robotic treatment of such tissue defects, not only in examining/reconstructing them. Would you please offer a better idea in the introduction why the scanning and reconstruction of wound would present a significant improvement in the therapy of such chronic diseases?

As mentioned in the manuscript conclusion:

“Development of fully automated medical station for chronic wound analysis will enable gathering of objective measurements about the status of the chronic wounds such as circumference, area, volume, tissue representation etc. The first step in obtaining those measurements in an automated fashion is the development of a robust robot system containing necessary hardware and software components needed for the precise 3D reconstruction of wound surface area.”

And, as also explained in the previous comment, development of a fully automatic recording system that will be able to enforce a certain surface sample density is only the first step in developing a larger system that will enable extrapolation of more concrete data such as wound’s physical parameters and percentile representation of specific tissue type (granulation, necrosis, fibrin). Such measurements and their progression over time will enable clinicians to track and apply appropriate therapy in a timely fashion. Furthermore, creating precise digital 3D reconstructions of patients wounds would facilitate collaboration between remote clinicians which could result in a deeper understanding of current state of patient’s wound and better therapy proposals.

In order to better present the developing system, this previous paragraph has been also added to the introduction section.

Round 2

Reviewer 1 Report

The authors revised their manuscript accordingly.